# TRecViT: A Recurrent Video Transformer

**Viorica Pătrăucean**[*1]                     *viorica@google.com*
**Xu Owen He**[*]                              *hexu@google.com*
**Joseph Heyward**[*]                          *heywardj@google.com*
**Chuhan Zhang**[*]                            *chuhanz@google.com*
**Mehdi S. M. Sajjadi**                        *msajjadi@google.com*
**George-Cristian Muraru**                     *gmuraru@google.com*
**Artem Zholus**                               *zholus@google.com*
**Mahdi Karami**                               *mahdika@google.com*
**Ross Goroshin**                              *goroshin@google.com*
**Yutian Chen**                                *yutianc@google.com*
**Simon Osindero**                             *osindero@google.com*
**João Carreira**                              *joaoluis@google.com*
**Razvan Pascanu**                             *razp@google.com*

*Google DeepMind*

**Reviewed on OpenReview:** *https://openreview.net/forum?id=Mmi46Ytb1H*

## Abstract

We propose a novel block for *causal* video modelling. It relies on a time–space–channel factorisation with dedicated blocks for each dimension: gated linear recurrent units (LRUs) perform information mixing over time, self-attention layers perform mixing over space, and MLPs over channels. The resulting architecture *TRecViT* is causal and shows strong performance on sparse and dense tasks, trained in supervised or self-supervised regimes, being the first causal video model in the state-space models family. Notably, our model outperforms or is on par with the popular (non-causal) ViViT-L model on large scale video datasets (SSv2, Kinetics400), while having 3× less parameters, 12× smaller memory footprint, and 5× lower FLOPs count than the full self-attention ViViT, with an inference throughput of about 300 frames per second, running comfortably in real-time. When compared with causal transformer-based models (TSM, RViT) and other recurrent models like LSTM, TRecViT obtains state-of-the-art results on the challenging SSv2 dataset. Code and checkpoints are available online `https://github.com/google-deepmind/trecvit`.

## 1 Introduction

Video understanding requires low-level scene understanding (e.g. how objects move) and high-level reasoning (e.g. causal relations between events) over a signal that is high-dimensional, can be noisy, and contains high correlations and redundancies in both spatial and temporal dimensions. Efficient video modelling needs high-capacity models that can represent the sheer diversity and richness of real-world videos, while operating in a causal manner with reasonable compute and memory footprint both at training and during inference time – these efficiency requirements are critical for Robotics or augmented reality applications. Convolutional neural networks (Carreira & Zisserman, 2017b; Feichtenhofer et al., 2019; Lin et al., 2019; Kwon et al., 2020) have been a successful family of causal models for video, but their scaling capabilities (in both data and parameters) are limited due to their inductive biases (locality, invariance). Recurrent neural networks, e.g. (Srivastava et al., 2015; Patraucean et al., 2016) have some desirable properties for video modelling (constant inference cost per timestep independent of the length of the video, causality for unidirectional

---

[*]Core contributor. [1]Corresponding author.

models), but they are slow to train due to their sequential nature and have difficulties in learning over long complex sequences. Transformers (Vaswani et al., 2017) have emerged as a very powerful family of models for all modalities, with impressive scaling capabilities. However, they have a significant memory footprint and latency due to the quadratic complexity of the self-attention operation and their performance degrades when using causal self-attention masks. Recently, a new family of linear recurrent networks (Gu et al., 2020; Gu & Dao, 2023; Orvieto et al., 2023b; Beck et al., 2024), referred to as State Space Models (SSMs), has emerged as an answer to the quadratic complexity of self-attention and the slow training of RNNs, with promising results for vision and language (De et al., 2024; Li et al., 2024). However, none of the existing video SSM architectures can run in a causal manner, their performance strongly depends on bidirectional operation.

In this paper, we propose a hybrid architecture that combines the best of all worlds. It alternates gated linear recurrent units (LRUs) (De et al., 2024) applied over time, with self-attention blocks over space, and MLP over feature channels. As opposed to space and channels, time has a natural order (*"arrow-of-time"*) that LRUs can implicitly and efficiently model in a causal manner with $O(N)$ complexity in the number of input frames at training time and $O(1)$ complexity at inference time, making it possible to process in real-time videos that extend even indefinitely. Space, on the other hand, has a fixed limited dimension, for which the quadratic cost of self-attention is more accessible. From a practical perspective, using self-attention over space allows us to naturally process in parallel all the pixels of a given frame, without having to commit to a particular scanning order (Li et al., 2024), making better use of hardware when parallel resources are available. Importantly, by restricting the LRUs to temporal-only recurrence, this factorisation reduces the sequence length by about two orders of magnitude compared to models that apply recurrence across both space and time (Zhu et al., 2024; Li et al., 2024). Such models typically require bidirectional scanning to attain strong performance, preventing them from operating in a causal manner.

To further limit the self-attention cost, we use spatial patches as introduced in the successful ViT (Dosovitskiy et al., 2021) model. But, compared to existing video transformer models, e.g. ViViT (Arnab et al., 2021), the patches do not have a fixed temporal extent. Instead, the embeddings of the spatial patches are integrated continuously into the hidden state of the gated LRUs, providing *persistent* memory of the entire temporal sequence up to the current frame, leading to a *causal* architecture. Furthermore, similar to convolutional networks, the parameters of the LRUs are shared over space, preventing the number of parameters from exploding as the resolution of the video increases.

We refer to the resulting model as *T*emporal *Rec*urrent *Vid*eo *T*ransformer (TRecViT). TRecViT is highly flexible and can address various video understanding tasks, both sparse (e.g. video classification) and dense (e.g. point tracking), trained in a supervised or self-supervised manner, e.g. using masked auto-encoding. In all our experiments, we use a causal setup that respects the arrow of time, so the model is suitable for any downstream applications, from e.g. video classification where we have offline access to the videos, to e.g. Robotics, where online processing is required. Overall, our model is significantly more efficient in both memory footprint and FLOPs compared to vanilla transformers, and obtains state-of-the-art performance when compared to causal models, while comfortably running in real time (e.g. throughput of about 300 frames per second for point tracking task).

**Contributions**: We propose a causal video architecture that uses a novel factorisation interleaving LRUs and ViT blocks. We run ablations for the building blocks to find optimal hyperparameters and present extensive experiments on multiple tasks and training regimes, showing the versatility and strong performance of our model.

## 2 Related work

**Transformers for Video.** Proposed initially as language models, transformers (Vaswani et al., 2017) have quickly become the dominant architecture across multiple modalities (images, audio, video). Transformer blocks alternate between a spatial mixing block represented by self-attention and a (feature) channel mixing block, typically represented by a gated MLP. Given that the self-attention layer treats the input tokens as *a set*, positional encodings must be used in order to specify the location of each token. This also implies that *no parsing* order is needed, unlike the case with RNNs. Vision transformers (ViT) (Dosovitskiy et al., 2021; Liu

et al., 2021) split images into a fixed number of patches that are projected into an embedding space to obtain *tokens* and these are then processed by a regular transformer. Several works extended ViT to video (Arnab et al., 2021; Bertasius et al., 2021; Li et al., 2021; Fan et al., 2021; Patrick et al., 2021), e.g. by replacing the regular image patches with spatio-temporal ones. The main challenge with transformers, particularly for video, is the quadratic complexity in the number of input tokens. Multiple approaches have been proposed to address this: e.g. factorisations of the self-attention operation (Arnab et al., 2021; Bertasius et al., 2021), iterative attention (Jaegle et al., 2022), sparse sampling of the input frames (Piergiovanni et al., 2023), and distributed self-attention operations across different devices (Liu et al., 2024). Our proposed model uses a novel space-time factorisation, where the temporal dimension is handled by LRUs and the spatial dimension by self-attention, resulting in a causal efficient video architecture.

As these models scale successfully to large number of parameters, their data needs are efficiently met by using self-supervised pre-training like masked autoencoding (MAE) (Tong et al., 2022) or contrastive learning (Zhao et al., 2024). Due to the factorisation used in our architecture, using such pre-training strategies is straightforward and we include successful experiments with MAE pre-training in Section 5.

**SSM, a type of Linear Recurrent Model.** While transformers (Vaswani et al., 2017) can be efficiently parallelised during training, at inference they need to pay a quadratic cost in the sequence length. On the other hand, recurrent networks (Elman, 1990; Hochreiter & Schmidhuber, 1997a; Mikolov et al., 2010; Bahdanau et al., 2014; Sutskever et al., 2014) are compact and efficient at inference but slow at training. State Space Models (SSMs) (Gu et al., 2020; 2021; Orvieto et al., 2023b), a particular type of linear recurrent networks, have recently been proposed as an answer to the scalability problem of RNNs, and have shown strong performance in language and other long-range dependencies tasks (De et al., 2024; Gu & Dao, 2023).

SSMs, like S4 (Gu et al., 2021), S4D (Gu et al., 2022), or Mamba (Gu & Dao, 2023) have been introduced as particular discretizations of a continuous time linear system. On the other hand, the linear recurrent unit (LRU) (Orvieto et al., 2023b) was designed by identifying the minimal set of changes to a vanilla RNN (Elman, 1990) that allows it to obtain the same key properties as the S4D architecture (Gu et al., 2021). Specifically, the nonlinear recurrence typical of a recurrent model was removed to improve the scalability and controllability of the system, as the linearity allows the recurrent matrix to be diagonalised through eigenvalue decomposition and absorbing the (dense) eigenvectors matrix into the neighbouring layers. This gives direct access to the eigenvalues of the Jacobian of the transfer function characterising the system. By parametrising the diagonal weight matrix containing the eigenvalues such that all entries are constrained to be below one, the system is guaranteed to be stable, bypassing exploding gradients issues. However, using only linear recurrence can greatly limit the expressivity of the layer. In (Orvieto et al., 2023a), the authors show that by using these layers within a typical transformer structure that alternates linear recurrences with point-wise nonlinearities (e.g. the MLP block), the overall architecture recovers expressivity through depth and can be shown to be a universal approximator of finite sequence-to-sequence maps.

Improving on the LRU, the gated LRU (De et al., 2024) introduces gating mechanisms similar to LSTM or GRU architectures, to filter the input sequence, or, for the recurrent gate, to control the rate of the information decay. Importantly, different from LSTM or GRU, these gates do not depend on the previous state, which would prevent parallelisation at training time. A series of recent works rely on formulating the linear recurrence as a form of linear attention, such as Mamba 2 (Dao & Gu, 2024), Gated Delta Networks (Yang et al., 2025), MesaNetworks (von Oswald et al., 2025), akin to previous observation, providing a more direct connection to attention. These works exploit this connection to show that the recurrence can be seen as an update rule of a local objective, similar to interpretability works on *in-context learning*. In this line of work, it is also worth noting xLSTMs (Beck et al., 2024), which connects SSMs more explicitly to traditional non-linear models such as LSTMs (Hochreiter & Schmidhuber, 1997b). In our work, we use gated LRUs, but we expect similar results when using other gated SSM blocks like Mamba or xLSTM within our factorisation.

**SSMs for Video.** While SSMs have mostly been explored in language, several architectures like S4 and Mamba have also been adapted to image and video understanding (Zhang et al., 2024) and generation (Gao et al., 2024). ViS4mer (Islam & Bertasius, 2022) uses a ViT image encoder to process videos frame by frame, and integrates their representations over time using S4 blocks at the top. TranS4mer (Islam et al., 2023)

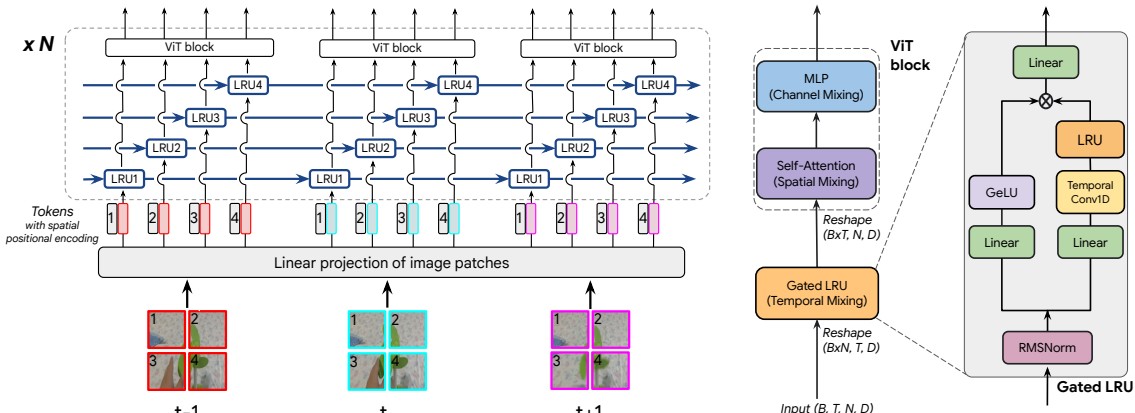

Figure 1: **Left:** TRecViT architecture. Each video frame is divided into non-overlapping patches that are linearly projected into a token embedding space. We then add a learnt spatial positional encoding. The tokens are passed through gated linear recurrent units (LRUs) that share parameters across space. The outputs of the recurrent blocks are then processed by a ViT block. The recurrent operation followed by ViT is repeated N times. **Right:** TRecViT block. The input is a batch of videos, each frame with N tokens. We apply recurrent units over *temporal tubes* to integrate information over time, and self-attention and MLP across tokens within each frame. Note that the recurrent units share parameters, but the information is not mixed across temporal tubes. Similarly, the ViT blocks share parameters, but the information is not mixed across frames. There are skip connections around each block, not included here to reduce clutter.

uses self-attention over short clips and integrates these with gated S4 blocks. More recently, the Mamba architecture was extended to images and videos by having it process a flattened 1D sequence of image or video patches. This requires defining a processing order for the patches, and different orders have been proposed, e.g. bidirectional and following a column or row order (Zhu et al., 2024; Li et al., 2024; Shi et al., 2025; Lu et al., 2025). As opposed to these Mamba-based architectures, our factorisation naturally uses the arrow-of-time to decide the scanning order, resulting in a *causal* model. Another important benefit of our hybrid architecture is that we can initialise the ViT blocks with strong existing pre-trained weights. This leads to strong performance even at larger scale, as opposed to VideoMamba (Li et al., 2024) where the authors report severe overfitting issues, requiring distillation from smaller models when training in a supervised fashion or distillation from CLIP features (Radford et al., 2021) for self-supervised training.

**Causal video models.** All the models mentioned above process the videos in an offline manner, i.e. require access to all frames at the same time, making their application in streaming setups (e.g. Robotics, virtual reality) sub-optimal. A few works have proposed adaptations of convnets and Transformers to causal operation (Lin et al., 2019; Zhao et al., 2023; Yang et al., 2022). In (Zhao et al., 2023), the authors store the keys and values from previous frames to perform cross attention with queries from the current frame, but they still use a non-causal temporal transformer for action recognition tasks. The closest to our work is RViT (Yang et al., 2022), which proposes a linear attention-based gating for processing videos in a recurrent manner. In the experiments section, we show that our recurrence based on LRUs leads to state-of-the-art results in causal operation settings and competitive performance with non-causal models.

## 3 TRecViT Architecture

**Notations:** Let $X \in [0, 1]^{T \times H \times W \times 3}$ be an RGB video with $T$ frames and $H \times W$ pixels. The video frames are split into $N$ non-overlapping patches $p_t^k$ of size $n \times n \times 3$, with $t \in \{1, T\}$ and $k \in \{1, N\}$. Let $x_t^k$ be the tokens obtained after the linear projection of the patches and the addition of the spatial positional encoding, with token size $1 \times 1 \times d$, where $d$ is the token feature dimension. A *temporal tube* is a sequence $\{x_t^k | t = \overline{1, T}\}$ containing the tokens from the same spatial location across frames.

The proposed architecture, TRecViT, is composed of repeated identical blocks, each performing a sequence of information mixing steps across the different dimensions of the video signal: time, space, and channels; see Figure 1. The mixing over the time dimension is handled by gated linear recurrent units (LRUs), similar to the one introduced in (De et al., 2024) for language. Each spatial token is associated with an LRU that processes the tokens within the same temporal tube over time, without mixing the information across temporal tubes, so each LRU has its own state – we use LRU1, LRU2,... in Figure 1 to highlight this. The LRUs share parameters over space, similar to a convolutional network. When applying this temporal mixing operation, the space dimension is transposed into the batch dimension.

The mixing over spatial and channel dimensions is handled by a standard ViT block, which first performs the spatial mixing through a self-attention operation, then the channel mixing by using an MLP. When performing the spatial and channel mixing, the time dimension is transposed into the batch dimension. We first perform temporal mixing, followed by spatial and channel mixing. We found this time-space order to produce better results compared to space-time order. We hypothesise that, by applying LRUs first, this allows them to focus on more local, easier to model, information at the first layer, instead of operating directly on features that mix information across the entire frame.

Empirically, we show that this time-space-channel factorization and choice of building blocks is more efficient for understanding temporal dynamics compared to video transformer approaches (e.g. ViViT (Arnab et al., 2021)) or pure SSM models. By applying self-attention over the spatial dimensions, we allow all tokens to attend to all the other tokens in parallel, without having to commit to a particular order (unlike in VideoMamba). We employ strong transformer blocks from ViTs for this operation, including their Imagenet pre-trained weights. The recurrence of the temporal processing enables efficient frame-by-frame inference over long videos, with constant memory footprint and causal operation.

## 3.1 Gated LRUs for Video

We adopt the gated variant of the LRU (De et al., 2024) to design our proposed block for video modelling. Although LRUs have been proposed for language, we hypothesise that they can effectively model video signals, given the continuous nature of the video and LRUs' inspiration from time continuous systems. We run extensive analysis and ablations for the different components and hyperparameters of the gated LRU block to find a configuration that works well for video; see analysis below and ablations in section 5.1.

The gated LRU operation is described by the below equations:

$$
\begin{align}
i_t &= \sigma(W_x x_t + b_x) \quad &\textit{input gate} \tag{1}\\
r_t &= \sigma\left(W_\lambda x_t + b_\lambda\right) \quad &\textit{recurrence gate} \tag{2}\\
\lambda_t &= \sigma(\lambda)^{\mathrm{C} \cdot r_t} \tag{3}\\
h_t &= \lambda_t \odot h_{t-1} + \sqrt{1 - \lambda_t^2} \odot (i_t \odot x_t) \tag{4}
\end{align}
$$

where $h_t \in \mathbb{R}^d$ is the state of the LRU, $\lambda_t \in \mathbb{R}^d$ is a vector containing the eigenvalues of the (diagonal) recurrence matrix[1], $i_t \in \mathbb{R}^d$ is the input gate controlling whether $x_t \in \mathbb{R}^d$ is integrated within the state $h_t$ of the LRU or not, and $r_t \in \mathbb{R}^d$ is the recurrence gate. The weights and biases of the LRU ($W_x \in \mathbb{R}^{d \times d}$, $W_\lambda \in \mathbb{R}^{d \times d}$, $b_x \in \mathbb{R}^d$, $b_\lambda \in \mathbb{R}^d$) are initialized using LeCun init (LeCun et al., 2012).

The (learnable) recurrence weights $\lambda$ are passed through a sigmoid function to ensure they are between 0 and 1, and are initialised such that $\sigma(\lambda)$ is sampled uniformly in $[\lambda_{\min}, \lambda_{\max}]$. These recurrent weights are raised to the power $\mathrm{C} \cdot r_t$, which effectively acts as a *gate* controlled by $r_t$ ( equation equation 2). $r_t$ is defined as a linear projection, with parameters $W_\lambda$ and $b_\lambda$, followed by a sigmoid function to ensure again the range $[0, 1]$. By raising element-wise $\sigma(\lambda)$ to $r_t$, the effective recurrence weight at some position $j$ can change between the $j$-th entry of $\sigma(\lambda)$ when the corresponding gate entry is 1 and 1 when the gate entry is 0.

---

[1]Similar to (De et al., 2024), we implement the recurrence weights $\lambda_t$ as $\exp(-C \cdot \mathrm{softplus}(\lambda) \cdot r_t)$, which is mathematically equivalent but numerically more stable.

The additional constant coefficient $C \in \mathbb{R}$, typically set to 8 as in (De et al., 2024), increases the range to be between $\sigma(\lambda)^C$ to 1, providing additional flexibility. E.g. if $\sigma(\lambda)$ is 0.9 and we set C = 8, we extend the range from $[0.9, 1]$ to $[0.43, 1]$. More importantly, we change the learning dynamics (e.g. gradient norms) and resolution we have over the range during learning. Specifically, for $x_t$ in some fixed interval and similar magnitude $W_\lambda$, as it is the case at initialisation, a higher value of C implies $\lambda_t$ will concentrate more towards the edges of the range. Note also that this is the dynamic range in which the recurrent weights can vary during inference as a function of the input tokens.

In (De et al., 2024), the authors found that setting $\lambda_{\min} = 0.9$ and $\lambda_{\max} = 0.999$ leads to the best results. An eigenvalue of 0.9 implies that it will take at least 10 time steps for the information to decay to roughly 35% of its magnitude, while for an eigenvalue of 0.999 it will take 1000 time steps to decay by the same amount. When using the same range for video modelling, we observed that the eigenvalues are pushed significantly towards $\lambda_{\min}$ during training, with a small number of eigenvalues becoming smaller than $\lambda_{\min}$; see Figure 2. We experimented with extending the range and obtained better results with $\lambda_{\min} = 0.6$; see Table 1. This leads to faster decay of information initially and might reflect the importance for videos of having enough recurrent units focused on short term information, in order to disentangle fast changing dynamics from slow ones.

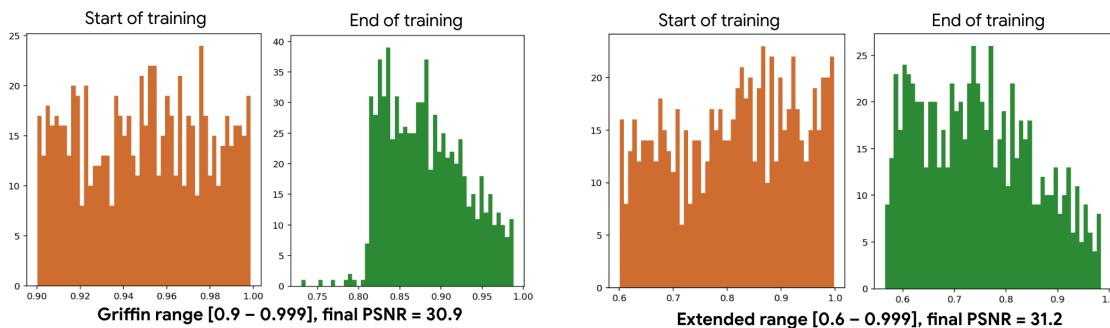

Figure 2: Distribution of the eigenvalues of the recurrent matrix at the beginning and end of training on long video memorisation task (see subsection 5.4) for different initialisation ranges.

Finally, note that when diagonalising the recurrence matrix, the eigenvalues $\lambda$ could, in theory, have complex values. We conducted experiments using complex eigenvalues, but we did not see improvements compared to using only real eigenvalues. The same observation was made in (De et al., 2024; Gu & Dao, 2023) as well.

## 3.2 Video block based on gated LRU

We use the gated LRU in a similar block structure as the one employed in (De et al., 2024), see Figure 1b. Given a 1D input (temporal tube), the block first applies a normalisation layer, then the signal is routed on two different paths. On the first one, it gets linearly projected to same dimensionality $d$ and then the *GeLU* activation is applied. On the other path, the signal is also linearly projected to the same dimensionality $d$, then we apply a 1D convolution followed by the gated LRU described in equation 4. The output of the LRU and the GeLU branch are element-wise multiplied and then linearly projected to the same dimension $d$. Note that, in line with (De et al., 2024), we use a separable convolution, which allows mixing information only over time, not over channels. We sweep the width of the convolutional kernel and find that a window of 4 gives the best results, similar to (De et al., 2024). Different from (De et al., 2024), we do not use an MLP block after the LRU for feature mixing. We apply the MLP after the self-attention block, as done in ViT.

Given the diagonal form of the recurrence, on device, the gated LRU computations are memory-bound, i.e. the data transfer takes longer than the actual computations done on that data. Similar to (De et al., 2024) we use a specialised *Pallas* (Bradbury et al., 2018) kernel that minimizes the number of bytes that need to be moved between HBM and VMEM (the Vector Processing Unit's cache). The parameters added by the linear projections within the block, as well as the parameters of the convolution and the LRU, are learned.

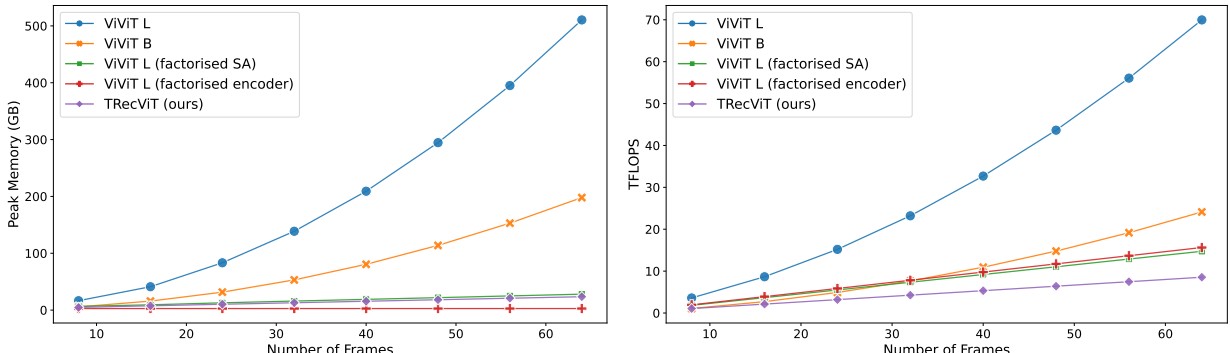

Figure 3: **Left:** Memory comparison; **Right:** FLOPs comparison. Our model demonstrates increasingly greater memory and compute savings compared to ViViT full self-attention as the number of frames increases. The efficiency of ViViT factorised self-attention is very similar to TRecViT's, but its performance is significantly lower. The factorised encoder variant of ViViT-L is very efficient, but it performs well mainly on sparse tasks.

## 4 Training TRecViT

The proposed architecture can be trained in a supervised or self-supervised regime. Given a tokenised video input, the output of TRecViT will have the same dimension and shape as the input, meaning that we can easily recover the spatio-temporal structure of the input video, which can be useful for dense tasks like pixel reconstruction, depth estimation, or point tracking. At inference time, the architecture can be applied over all the video frames at once, or frame-by-frame by carrying over the state of the LRUs. Depending on the task, one can choose to keep all the outputs from all time-steps to make a prediction (similar to ViViT), or just the outputs from the last step, given that the LRU integrates the previous history in its state. In our experiments, we use mainly the former for fairer comparison with ViViT, but we also experiment with the latter to analyse LRU's capability of remembering over a very long context; see subsection 5.4.

### 4.1 Self-supervised pre-training

Given the factorised nature of the proposed architecture and the redundancy present in the video signal, it comes natural to apply masked auto-encoding to enable self-supervised pre-training from scratch on large-scale unlabelled datasets.

We follow the same recipe as in the original VideoMAE paper (Tong et al., 2022). Specifically, we use tube masking where a 2D random mask is generated and repeated for all the frames in the video. For our architecture, this is equivalent to dropping temporal LRUs. The training objective is simply $L_2$ reconstruction error of the entire frames. We sweep the value of the masking ratio and we find that 0.90 leads to best performance on downstream tasks. When using the pre-trained representations for downstream tasks, we keep all the tokens of the video and we add a decoder or readout head that is fine-tuned for the respective tasks.

### 4.2 Computational complexity and efficiency analysis

We discuss here the computational complexity of the proposed architecture and its efficiency compared to different variants of ViViT. Our model's parameter count (111M) falls between that of ViViT-B (90M) and ViViT-L (310M). We focus on ViViT-L as the main point of comparison in our efficiency and SOTA analysis (Section 5.2) as this is the strongest representative of the ViViT family reported in the original ViViT paper. Furthermore, since our architecture is suitable for both sparse and dense video tasks, we compare the efficiency benefits mainly against the more general full self-attention ViViT-L variant, which is utilised by influential follow-up works (e.g., VideoMAE Tong et al. (2022)) over the specialised Factorised Encoder (FE) version, which models temporally only one token per frame, being suitable for classification tasks, but not for dense tasks.

We consider a video with $T$ frames, each frame with $N$ spatial tokens of dimension $D$, and a model with $L$ layers. The full self-attention ViViT flattens the video sequence and computes attention scores between all $T \cdot N$ pairs of tokens followed by a token-wise MLP block, resulting in a complexity $\mathcal{O}(LT^2N^2D + LTND^2)$, with the first term associated to self-attention dominating the overall computation. For our hybrid architecture, we consider the complexity of the ViT blocks and the gated LRU blocks separately. The $L$ ViT blocks perform spatial attention over the $N$ tokens within each frame, resulting in $\mathcal{O}(LT(N^2D + ND^2))$ complexity. The $L$ gated LRU blocks perform temporal processing across the $T$ frames for all $N$ tokens in parallel, contributing $\mathcal{O}(LTND^2)$ complexity. Hence the total complexity for the proposed model is $\mathcal{O}(LTN^2D + LTND^2)$. The main difference is that ViViT full self-attention scales quadratically with the number of frames ($T^2$) and the number of spatial tokens ($N^2$), whereas our model scales linearly with the number of frames ($T$) and quadratically with the number of spatial tokens ($N^2$).

This difference is reflected in the memory footprint and FLOPs count of the two types of models; see Figure 3. The profiling results are obtained by cost and memory analysis of lowered Jax HLO on CPU backend to be aligned with the theoretical numbers (JaxAuthors, 2018). We consider as input a video of size $224 \times 224$ and we vary the length of the video to analyse the savings provided by our architecture as the length of the video increases. The peak memory and number of flops for TRecViT are significantly lower as the number of frames increases, e.g. at 32 frames (the number of frames typically used in video classification experiments), TRecViT's peak memory is $\sim 12\times$ smaller than that of ViViT-L full self-attention and the FLOPs count is $5\times$ lower. When going to 64 frames, the peak memory is $\sim 24\times$ smaller and FLOPs count is $8\times$ lower. This is due to the fact that TRecViT maintains its history in a fixed-sized compressed state, whereas full self-attention Transformer-based models store in cache the keys and values for the full history. Similarly, the FLOPs count grows exponentially for ViViT due to its quadratic complexity, where each input token attends to every other token in the video. The factorised self-attention and factorised encoder variants are significantly more efficient than the full self-attention and comparable to TRecViT. However, the performance of the factorised self-attention version is significantly worse (see ablations in Table 2), whereas the factorised encoder is tailored specifically for sparse tasks, modelling only one class token per frame.

## 5 Experiments

We present results for supervised video classification and self-supervised masked auto-encoding with frozen representations evaluated on two downstream tasks: video classification and point tracking. To analyse the memory capabilities of our model, we also include a reconstruction task of frames seen in the distant past. Using the same task, we study the generalisation capabilities to longer sequences than seen during training. We follow the ViT scaling configurations and, unless otherwise stated, we use the **B**ase version for our model for all our experiments. We specify the number of parameters for all models considered in our experiments, and we include in the supplementary material all the model configurations, training hyperparameters, and data augmentations used in all experiments.

**Datasets:** We use mainly two large-scale real-world datasets for ablations and for SOTA comparison on the supervised video classification task. Kinetics400 (Carreira & Zisserman, 2017a) contains 241,512 videos[2] across train, validation, and test splits, 10s-long (25fps), spanning 400 classes. This dataset is known to require modelling appearance for successful action recognition. To challenge our model's capability of understanding motion, we also use SSv2 dataset (Goyal et al., 2017), which contains 220,847 shorter videos (2-6s long), sampled at 12fps, representing 174 classes. This dataset includes actions that differ in finer motion-related details, requiring a deeper temporal understanding, e.g. *pouring something into something* vs *pretending to pour something into something.*

### 5.1 Ablations

**Gated LRU block:** We ran multiple ablations for the components of the gated LRU block used in TRecViT using SSv2 supervised classification as task, see Table 1. We study the impact of the different components and

---

[2]Kinetics is a dynamic dataset (videos may be removed from YouTube). Our current version has 241,512 videos, compared to 267,000 videos reported in (Arnab et al., 2021), so a decrease of almost 10%, noticeable in the final performance.

of the initialisation range for the eigenvalues of the recurrent matrix, as mentioned in section 3.1. We report top-1 classification accuracy, i.e. the percentage of videos for which the model's highest-confidence prediction matches the ground truth label. The results show that the skip connection is essential to successful training; without it, we need to lower the learning rate for the training to take off at all, and the final performance is still very poor. Removing the input and recurrent gates leads to slightly reduced performance, in line with the observations on SSM in language. The input gate adds selectivity on the input, while the recurrent gate modulates how far in the past the model looks. Removing the 1D convolutional layer leads to a bigger decrease, which again is in line with the original Griffin paper that mentioned the important role of the Conv1D layer in extracting local temporal features.

| Ablation | Top-1(%) |
|---|---|
| Gated LRU block [0.6, 0.999] (ours) | **66.7** |
| Griffin range [0.9, 0.999] | 66.2 |
| No skip connection (lower lr) | 15.5 |
| No input gate, no recurrent gate | 66.0 |
| No convolutional layer | 64.9 |

Table 1: Ablation on Gated LRU components and eigenvalue initialisation. Task: SSv2 classification. Metric: top-1 accuracy (higher is better).

**Training from scratch:** We run an experiment to compare the proposed factorisation against simpler adaptations of LRU to video, similar to the ones used in ViViT and VideoMamba, i.e. patchify and flatten the entire video, add spatio-temporal positional embeddings, and apply 1D self-attention or SSM blocks as done in language tasks. Figure 4 compares the proposed TRecViT with these two baselines, with all models being trained from scratch on supervised classification on SSv2. We consider the **S**mall version for all models as the larger **B**ase version shows stability issues when trained from scratch, as reported in other works as well (Li et al., 2024; Arnab et al., 2021). As expected, the performance on this challenging dataset when training from scratch is far from SOTA, but it clearly shows that the proposed factorisation has superior video modelling capabilities compared to baselines, ViViT-S with full self-attention being the closest competitor. PureLRU's performance is very poor, which is in line with the findings from VideoMamba that bidirectional (non-causal) processing of the input is needed for good performance.

**Same factorisation, different temporal blocks:** We compare our proposed hybrid architecture against other factorised variations that have similar efficiency; see Figure 5 in the appendix for memory footprint and FLOPs comparison. The results are included in Table 2 and clearly show that the proposed hybrid architecture is superior in performance and in training throughput compared to architectures using LSTM or factorised self-attention to integrate temporal information. The model using Conv1D as temporal module is faster to train, but its accuracy is far from competitive.

**Additional ablations:** We include in the appendix ablations by running supervised classification on SSv2 using different hyperparameter values: temporal dimension of video patches, minimal radius for eigenvalue initialisation, window size for the 1D convolution in LRU. We also include an experiment considering multiple seeds and report mean and variance.

| Temporal module (w skip) | Top-1 (%) | Steps per sec (training) |
|---|---|---|
| ViViT factorised self-attention | 62.4 | 8.3 |
| ViT-LSTM | 63.7 | 6.0 |
| ViT-Conv1D | 40.3 | 10.5 |
| TRecViT | **66.7** | 8.9 |

Table 2: Comparison of different temporal modules used in hybrid setups similar to our proposed TRecViT. Task: SSv2 classification. Metric: Top-1 accuracy (higher is better).

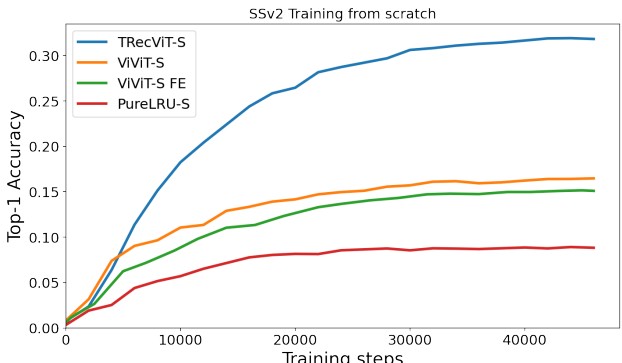

Figure 4: TRecViT compared to baselines on supervised video classification on SSv2 dataset, trained from scratch. The plot shows the evolution of the evaluation accuracy as training progresses (higher is better).

## 5.2 SOTA comparison on supervised video classification

We compare the performance of TRecViT against top models in the literature, considering both causal and non-causal architectures; see Table 3 for SSv2 results and Table 4 for Kinetics400 results. On SSv2, our model achieves state-of-the-art performance compared to causal baselines (TSM, RViT, causal ViViT) and outperforms or is competitive with all non-causal baselines. Notably, it outperforms the popular ViViT-L by 2.3% despite having about 3× less parameters.

On Kinetics400, TRecViT outperforms convolutional architectures (I3D) and some Transformer-based architectures (TimeSformer, ViViT-B, causal ViViT-L). Compared to RViT and ViViT-L, the performance is competitive, but slightly lower. We attribute this in part to the reduced number of videos in the dataset as mentioned above – when training both TRecViT and ViViT-L on the same number of videos, the performance gap is significantly reduced. In addition, this result could also reflect the difference between the two datasets mentioned above (appearance vs motion) and highlights that TRecViT is superior at modelling motion compared to baselines, but on Kinetics400 where the appearance is enough for successful classification, the performance is on par.

## 5.3 Self-supervised masked autoencoding

We use Kinetics400 for self-supervised pre-training from scratch and we report results on multiple downstream datasets and tasks by fine-tuning attention readout heads on top of frozen representations. We choose this setup, as opposed to fine-tuning end-to-end, as the performance in this case more clearly reflects the quality of the pre-trained representations. As mentioned in the previous section, we use a large masking ratio (0.90), which makes pre-training very efficient. We report the number of parameters for every model considered. Note that the number of parameters for TRecViT is different from the one reported in the previous section due to the addition of the readout heads.

**Video classification:** We report video classification accuracy as downstream task using attention readout heads on SSv2 and Kinetics400. We compare the performance against VideoMAE-L (Tong et al., 2022) in Table 5. Our model obtains slightly better performance on both datasets compared to this strong baseline, despite having almost 3× less parameters.

**Point tracking:** To demonstrate that our model can handle dense(r) tasks as well, we evaluate the same frozen MAE representations for the point tracking task. We use the recurrent architecture in MooG (van Steenkiste et al., 2024) as a readout head due to its simplicity. MooG uses light cross-attention layers to process the embeddings of each frame in order, and the readout state is carried over time. We finetune the readout head using MOVi-E dataset (Greff et al., 2022) as done in popular point tracking works (Doersch et al., 2023). We evaluate these fine-tuned representations on two datasets: Perception Test (Pătrăucean et al., 2023) and DAVIS dataset (Pont-Tuset et al., 2017) with point tracks extracted in (Doersch et al., 2022). We report average Jaccard metric, i.e. the percentage of predicted points that fall within a certain threshold

| Model | Pre-Train | Top-1 (%) | Param (M) | FLOPs (T) | Mem (G) |
|---|---|---|---|---|---|
| *non-causal* | | | | | |
| I3D | K400 | 51.3 | 25.0 | N/A | N/A |
| SlowFast R50 | K400 | 61.9 | 34.1 | 0.19 | 3.35 |
| SlowFast R101 | K400 | 63.1 | 53.3 | 0.32 | 4.20 |
| MSNet | IN-21K | 64.7 | 54.6 | 0.07 | 6.54 |
| TimeSformer-L | IN-21K | 62.4 | 121.4 | 5.1 | >24 |
| VidTr-L | - | 63.0 | N/A | 10.5 | N/A |
| ViViT-L$_{32}$ | - | 65.9 | 310.8 | 7.75 | 6.11 |
| Mformer-B | IN-21K | 66.5 | 114.0 | 1.10 | 7.3 |
| MViT-B$_{32}$ | K400 | 67.1 | 36.6 | 0.51 | 10.7 |
| MViT-B$_{64}$ | K400 | 67.7 | 36.6 | 1.36 | >24 |
| MViT-B-24$_{32}$ | K600 | 68.7 | 36.6 | 1.36 | >24 |
| VideoMamba | IN-21K | 68.4 | 74.0 | 0.33 | N/A |
| VideoMambaPro | IN-21K | **69.4** | 72.0 | 2.2 | N/A |
| *causal* | | | | | |
| TSM* | K400 | 63.3 | 42.9 | 0.19 | 5.98 |
| cViViT-L$_{32}$ | IN-21K | 64.4 | 310.8 | N/A | N/A |
| RViT-L$_{32}$ | K400 | 66.1 | 72.0 | 1.34 | 2.12 |
| RVIT-XL$_{64}$ | K400 | 67.9 | 107.7 | 3.99 | 2.33 |
| *TRecViT*$_{32}$ | IN-21K | 66.8 | 111.3 | 1.44 | 1.79 |
| *TRecViT*$_{64}$ | IN-21K | **68.2** | 111.3 | 2.89 | 3.16** |

Table 3: Performance comparison on SSv2 dataset, considering causal and non-causal baselines. cViViT-L is a causal ViViT model that we trained using causal attention masking for the self-attention block. We do not include FLOPs and Mem for this model as they will heavily depend on the implementation of the causal masking. Where present, the subscript indicates the number of frames used at inference time. Metric: Top-1 accuracy (higher is better). *Result reported in (Yang et al., 2022). **For TRecViT, the memory at inference is independent of the number of frames, here we run causally on the entire sequence in parallel, hence the memory increase.

| Model | Pre-Train | Top-1 (%) | Param (M) |
|---|---|---|---|
| *non-causal* | | | |
| I3D | IN-1K | 72.1 | 25.0 |
| TimeSformer | IN-21K | 78.0 | 121.4 |
| Mformer-B | IN-21K | 79.7 | 114.0 |
| ViViT-L | IN-21K | 80.3 | 320.0 |
| ViViT-B | IN-21K | 78.1 | 91.2 |
| ViViT-L | IN-21K | 78.7 | 310.9 |
| *causal* | | | |
| RViT-XL | IN-21K | **81.5** | 107.7 |
| cViViT-L | IN-21K | 76.3 | 310.9 |
| *TRecViT* | IN-21K | 76.5 | 111.2 |

Table 4: Performance of TRecViT compared to convolutional and transformer-based baselines on Kinetics400 dataset. For fair comparison, we trained ViViT-B and ViViT-L models on the current Kinetics400 dataset version; see footnote 2. The numbers in gray correspond to results obtained on the original larger Kinetics400 dataset, reported by their authors. Metric: Top-1 accuracy (higher is better).

from the ground truth points (Doersch et al., 2022), for TRecViT compared with MooG and VideoMAE; see Table 7. TRecViT obtains better performance on both datasets compared to baselines, which reinforces the observation that our proposed model has strong motion modelling capabilities. We include qualitative results for this task in the appendix.

## 5.4 Long video memorisation task

Transformer models for language are known to be excellent at retrieving information from context, as they cache the keys and values for the entire history. On the other hand, LRUs / SSMs and RNNs in general

| Model | Dataset | Top-1 (%) | Param (M) |
|---|---|---|---|
| VideoMAE | Kinetics400 | 45.8 | 330 |
| *TRecViT* | Kinetics400 | **46.0** | 128 |
| VideoMAE | SSv2 | 53.7 | 330 |
| *TRecViT* | SSv2 | **53.9** | 128 |

Table 5: Performance of TRecViT compared to VideoMAE on video classification using frozen MAE representations, pre-trained on Kinetics400. Metric: Top-1 accuracy (higher is better).

| Model | Num frames train | Num frames test | PSNR |
|---|---|---|---|
| ViViT-L | 96 | 96 | **32.2** |
| *TRecViT* | 96 | 96 | 29.1 |
| ViViT-L | 64 | 96 | 15.1 |
| *TRecViT* | 64 | 96 | **26.4** |

Table 6: Comparison between TRecViT and ViViT-L on the long video memorisation task, when the models are evaluated on the same number of frames as seen in training (**top**) or on longer sequences (**bottom**). Metric: Peak Signal-To-Noise Ratio (PSNR); higher is better.

struggle with such *needle-in-the-haystack* style tasks as they need to perform the retrieval based on the compressed history kept in their recurrent state (Jelassi et al., 2024; De et al., 2024).

We are interested in studying this aspect in the video domain as well. We set up a simple reconstruction task where the model has to remember the frame seen at a given time-step in the past. For example, we train the models with sequences of length $T = 96$ frames to reconstruct the 16th frame. Using the same task, we also analyse the generalisation capabilities to sequences longer than those seen during training, i.e. the models are trained with sequences of length $T = 64$ frames to reconstruct the 16th frame, and then are evaluated on video sequences with $T = 96$ frames to reconstruct the same 16th frame.

We employ Walking Tours dataset (Venkataramanan et al., 2024), which contains hour-long videos, and the scenery changes constantly, hence we are guaranteed that the video frames seen most recently will be very different compared to the frames seen earlier on. We scale the videos to $224 \times 224$ pixels. We adopt ViViT-L as baseline, and we train both models using Imagenet pretrained weights. For ViViT-L, we keep all the outputs from all $T$ time steps and apply temporal pooling and a $1 \times 1$ convolution to get the expected shape for the reconstructed frame. For TRecViT, we simply keep the output of the last layer at time step $T$ and reshape it to the expected shape. When evaluating on longer sequences, the ViViT model needs to adapt the positional encoding – we use interpolation to nearest neighbour to obtain the desired length; cubic interpolation led to worse results. TRecViT can run on any sequence length without modification.

Table 6 includes quantitative results measured in terms of Peak Signal-to-Noise Ratio (PSNR), which is defined based on the mean squared error (MSE) between the pixel values of the original video and the reconstructed video; more quantitative and qualitative results are included in the Appendix. When evaluated on the same sequence length, ViViT-L is better than TRecViT as it manages to better preserve the higher

| Model | Dataset | AJ | Param (M) |
|---|---|---|---|
| MooG | DAVIS | 0.687 | 35 |
| VideoMAE | DAVIS | 0.703 | 330 |
| *TRecViT* | DAVIS | **0.706** | 128 |
| MooG | Perception Test | 0.760 | 46.5M |
| VideoMAE | Perception Test | 0.761 | 330M |
| *TRecViT* | Perception Test | **0.783** | 128M |

Table 7: Performance of TRecViT compared to baselines on point tracking task on DAVIS and Perception Test datasets. All models use frozen representations evaluated using the readout head from MooG. In line with MooG, we use 8 frames for the DAVIS dataset and 16 frames for the Perception Test. Metric: average Jaccard (AJ); higher is better.

frequencies in the visual signal. However, when evaluated on longer sequences, ViViT's PSNR drops significantly, showing strong artefacts, whilst TRecViT's output quality remains quite satisfactory (see qualitative results in the appendix, Figures 8 and 9). We consider this to be a very promising result that warrants further investigation into the memorisation capabilities of our model – we leave this as future work.

## 6  Conclusion

We propose a novel causal video architecture TRecViT that alternates gated linear recurrent units (LRUs) modelling the temporal dynamics in the video with ViT blocks modelling the spatial and channel dimensions. Notably, this is the first causal architecture in the SSM video models family. The proposed model outperforms or obtains competitive performance compared to strong causal and non-causal baselines on supervised and self-supervised tasks, while having a much smaller number of parameters and significantly reduced memory footprint and FLOPs count compared to vanilla Transformer video models. Our study indicates that temporal linear recurrence combined with spatial self-attention is a natural and effective parameterisation for video modelling, given the sequential nature of videos as defined by the arrow-of-time.

**Limitations and Future Work:** While this study presents the first exploration into leveraging LRUs for the video domain, some key aspects necessitate further investigation. First, we plan to conduct a deeper investigation into the memorisation limitations of our architecture, particularly in needle-in-haystack problems. Second, we aim to understand our model's capabilities in a multi-step generative context, specifically by integrating it within video diffusion models. Finally, given LRUs' inspiration from continuous time systems, we plan to explore its capabilities of modelling variable frame rate videos across various complex applications, including video-language tasks and Robotics.

### Acknowledgments

We would like to thank Çağlar Gülçehre, Jörg Bornschein, Zhitao Gong, Daniel Zoran, and Andrew Zisserman for providing insightful feedback on this work.

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
