We include here all the model configurations and hyperparameters used in the experiments presented in the main paper, together with ablations and qualitative visualisations of results for the point tracking task (section 5.3) and the long video memorisation task (section 5.4). Videos showing point tracks are also attached.

## A  Model configurations

Table 8 includes the model configurations used in our experiments.

| Model | Layers | Hidden size $D$ | ViT MLP size | ViT Heads |
|---|---|---|---|---|
| TRecViT-Small | 12 | 384 | 1536 | 6 |
| TRecViT-Base | 12 | 768 | 3072 | 12 |
| TRecViT-Large | 24 | 1024 | 4096 | 16 |

Table 8: Model Configurations used in our experiments

## B  Training hyperparameters

### B.1  Supervised video classification

| Hyperparameter | Kinetics400 | SSv2 |
|---|---|---|
| Peak learning rate | 1e-4 | 1e-4 |
| Weight decay | 0.03 | 0.03 |
| Label smoothing | 0.1 | 0.1 |
| Scale jitter | (0.875, 1.33) | (0.875, 1.33) |
| Num frames | 32 | 32, 64 |
| Stride | 2 | 2 |
| Cls dropout | - | 0.1 |
| Rand augment | - | yes |
| Epochs | 30 | 35 |
| Spatial crops eval | 3 | 3 |
| Temporal clips eval | 4 | 4 |

Table 9: Hyperparameter values used in the supervised classification experiments. These are mainly the hyperparameters used in previous works, e.g. ViViT Arnab et al. (2021). For both datasets, we use cosine decay for the learning rate schedule with linear warmup.

### B.2  Self-supervised masked autoencoding and fine-tuning

## C  Efficiency comparison against ViViT variants and other hybrid baselines

Figure 5 complements Figure 3 from the main paper and includes the memory footprint and FLOP counts for other hybrid baselines that use other types of temporal modules (Conv1D, LSTM) instead of the gated LRU used in our proposed architecture. As expected, these hybrid architectures scale linearly in the number of frames, similarly to our model. However, they are outperformed by our model, as mentioned in Table 2 in the main paper.

## D  Ablations

We include here ablations for different hyperparameters used in our model by running supervised classification experiments on SSv2. We sweep the following hyperparameters: temporal size of the video patches (Table 13),

| Hyperparameter | Kinetics400 |
|---|---|
| Learning rate | 3e-4 |
| Weight decay | 0.05 |
| Num frames | 16 |
| Stride | 2 |
| Epochs | 1600 |
| Mask ratio | 0.9 |

Table 10: Hyperparameter values used in the self-supervised masked auto-encoding experiment on Kinetics400. We use AdamW optimizer. We apply patch-wise normalisation of the inputs as done in Video-MAE Tong et al. (2022)

| Hyperparameter | Kinetics400 | SSv2 |
|---|---|---|
| Learning rate | 3e-4 | 3e-4 |
| Scale jitter | (0.9, 1.33) | (0.9, 1.33) |
| Num frames | 16 | 16 |
| Stride | 2 | 2 |
| Epochs | 30 | 6 |
| Spatial crops eval | 3 | 3 |
| Temporal clips eval | 4 | 4 |

Table 11: Hyperparameter values used in the fine-tuning classification experiments. We use cosine decay for the learning rate schedule with 1k steps of linear warmup.

| Hyperparameter | DAVIS | Perception Test |
|---|---|---|
| Learning rate | 3e-4 | 3e-4 |
| Num frames | 8 | 16 |
| Num steps | 200k | 40k |

Table 12: Hyperparameter values used in the point tracking fine-tuning experiments. We use cosine decay for the learning rate schedule with 1k steps of linear warmup.

| Patch temporal size | Top-1 (%) |
|---|---|
| 1 | 66.8 |
| 2 | 64.5 |
| 4 | 61.5 |
| 8 | 57.7 |

Table 13: Performance when using different temporal sizes for the video patches for supervised classification on SSv2, using 32 frames per clip. Our model performs best when the input is fed as spatial patches $t = 1$, with accuracy dropping significantly when using $t > 1$. We hypothesise that an increased temporal size leads to a less continuous signal fed into the LRUs, affecting its performance.

window of the 1D convolution kernel applied in LRU (Table 14), value of the minimal radius when initialising the eigenvalues of the recurrence matrix (Table 15). Finally, we run an experiment using five seeds on SSv2 classification using 32 frames, obtaining $66.6 \pm 0.2$; we include the best seed result (seed=0) in our SOTA comparison as done in other works as well (Yang et al., 2022; Arnab et al., 2021), and use this seed for the other experiments.

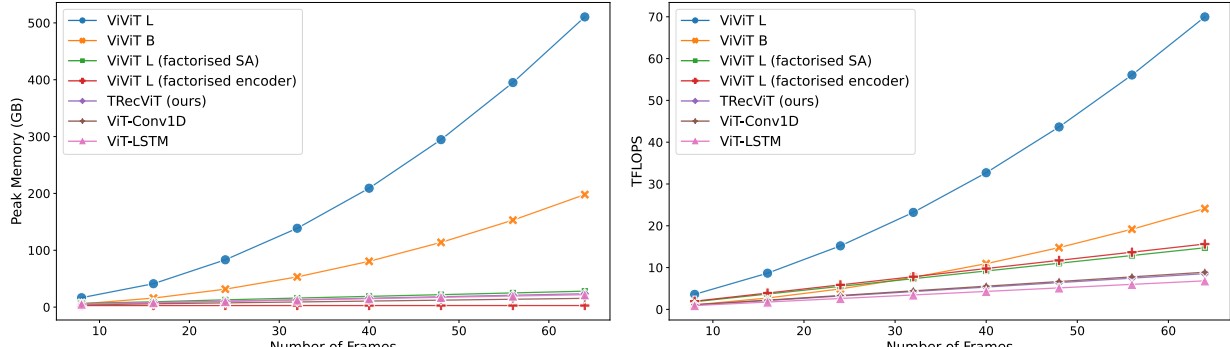

Figure 5: **Left:** Memory comparison; **Right:** FLOPs comparison. Our model has similar efficiency to other factorised architectures, like ViViT factorised self-attention, ViViT factorised encoder, ViT-Conv1D, ViT-LSTM, but it outperforms these baselines in accuracy and generality.

| Window size | Top-1 (%) |
|:---:|:---:|
| 2 | 66.4 |
| 4 | 66.8 |
| 8 | 65.7 |

Table 14: Performance when using different window sizes for the conv 1D kernel in the LRU, for supervised classification on SSv2, using 32 frames per clip. As found in Griffin as well, the best window size is 4.

| Min rad eigenvalues | Top-1 (%) |
|:---:|:---:|
| 0.6 | **66.8** |
| 0.7 | 66.6 |
| 0.8 | 66.5 |
| 0.9 | 66.2 |

Table 15: Performance when using different values for the minimal radius when initialising the eigenvalues of the recurrence matrix for supervised classification on SSv2, using 32 frames per clip. Compared to Griffin where $\lambda_{\min} = 0.9$ was found to give the best results, for video it is important to lower this value to 0.6, to allow for faster decay of information for some frequencies. We plan to conduct more investigations on this aspect to better understand the connection between $\lambda_{\min}$ and the temporal context of the task being performed.

# E  Point tracking qualitative results

In Figure 6, we include more visualisations for the point tracking task using frozen MAE representations pre-trained on Kinetics400, using TRecViT as backbone. Videos showing point tracks are also attached.

# F  Long video memorisation task

We run multiple experiments where the model is tasked to reconstruct the $(T-k)^{\text{th}}$ frame from the past, with increasing value for $k \in \{16, 48, 80, 112, 144, 164\}$ frames. For easier visual comparison, we increase the distance $k$ to the frame to reconstruct while also increasing the video length $T$, so the frame to reconstruct is always the same.

We show quantitative and qualitative results in Figure 7 and Figure 8, respectively. We can observe that there is a performance–efficiency trade-off at play for TRecViT: its performance is slightly below ViViT's for shorter memory spans (16, 48, 80), with the high frequencies being less well reconstructed as $k$ increases, but its efficiency (steps-per-second) is significantly higher. However, beyond 80 frames, ViViT-L goes out of

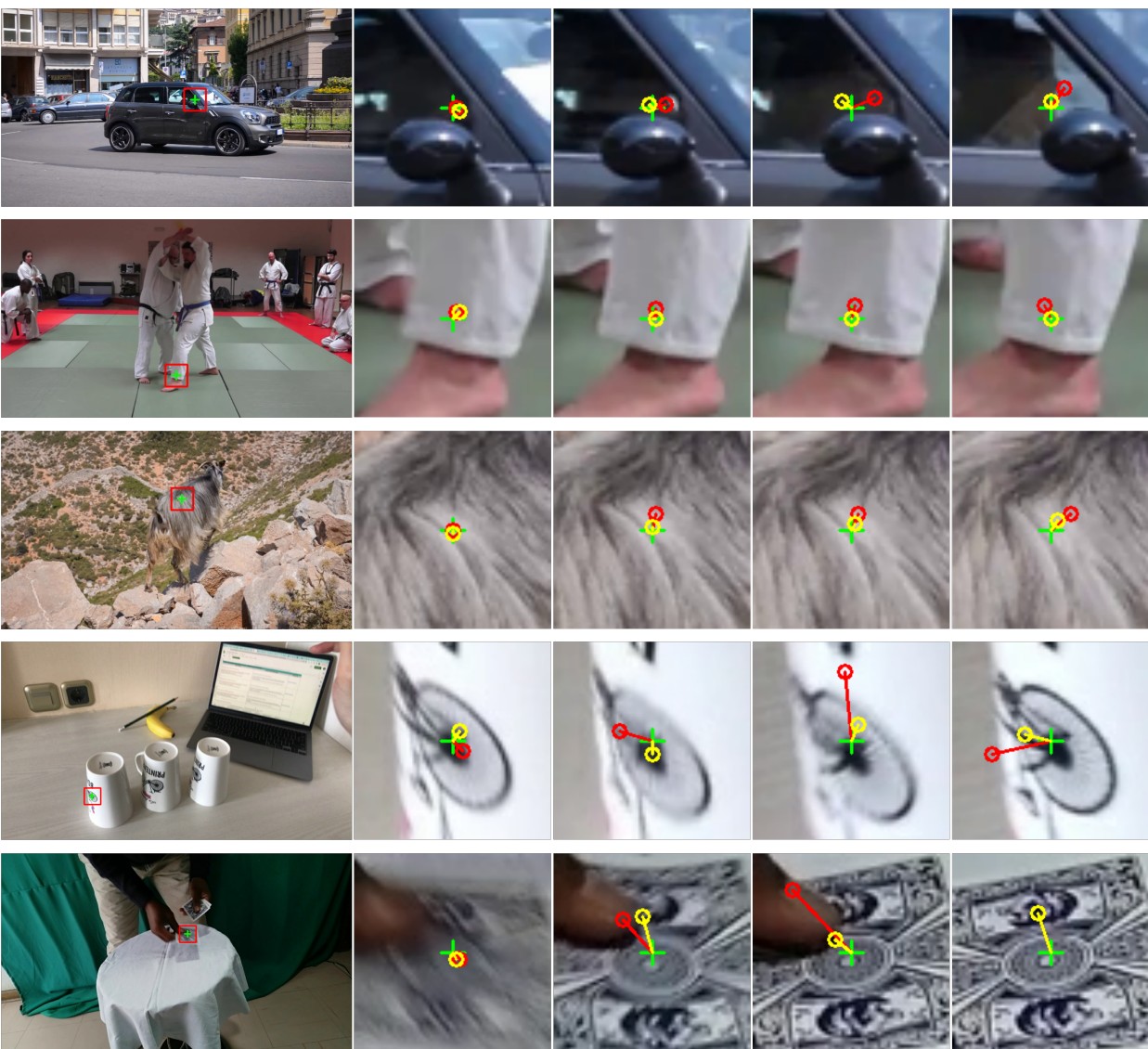

Figure 6: Qualitative results obtained by TRecViT for point tracking on DAVIS dataset (rows 1-2) and Perception Test (rows 3-4) compared to VideoMAE. The leftmost image indicates the point to track in the original frame, and the images towards the right show zoom-ins on subsequent frames. Green plus (+) marker indicates the ground truth, yellow circle indicates TRecViT's predictions and red circles indicate VideoMAE's predictions.

memory being unusable, whilst TRecViT continues to give decent results up to $T = 160, k = 144$, i.e. it is able to learn with sequences of up to 5.3s long at 30FPS, and remember a frame seen about 4.8s before.

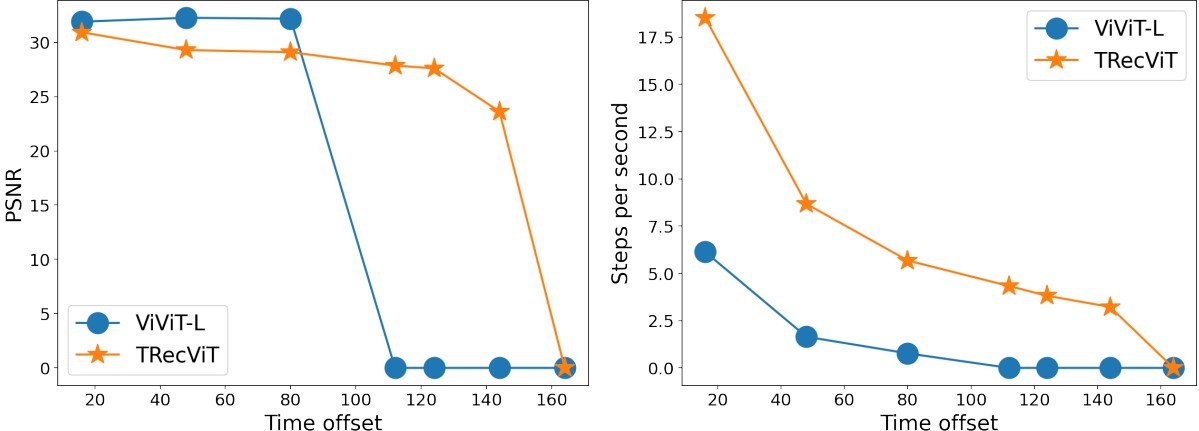

Figure 7: Long video memorisation task. **Left:** PSNR comparison; **Right:** Step-per-second comparison. At time $T$, the model has to reconstruct the $(T - k)^{\text{th}}$ frame seen in the past. The plots show PSNR and throughput (steps-per-second) for increasing time offset $k$. For both models, the data points with 0 value on the $y$-axis correspond to OOM.

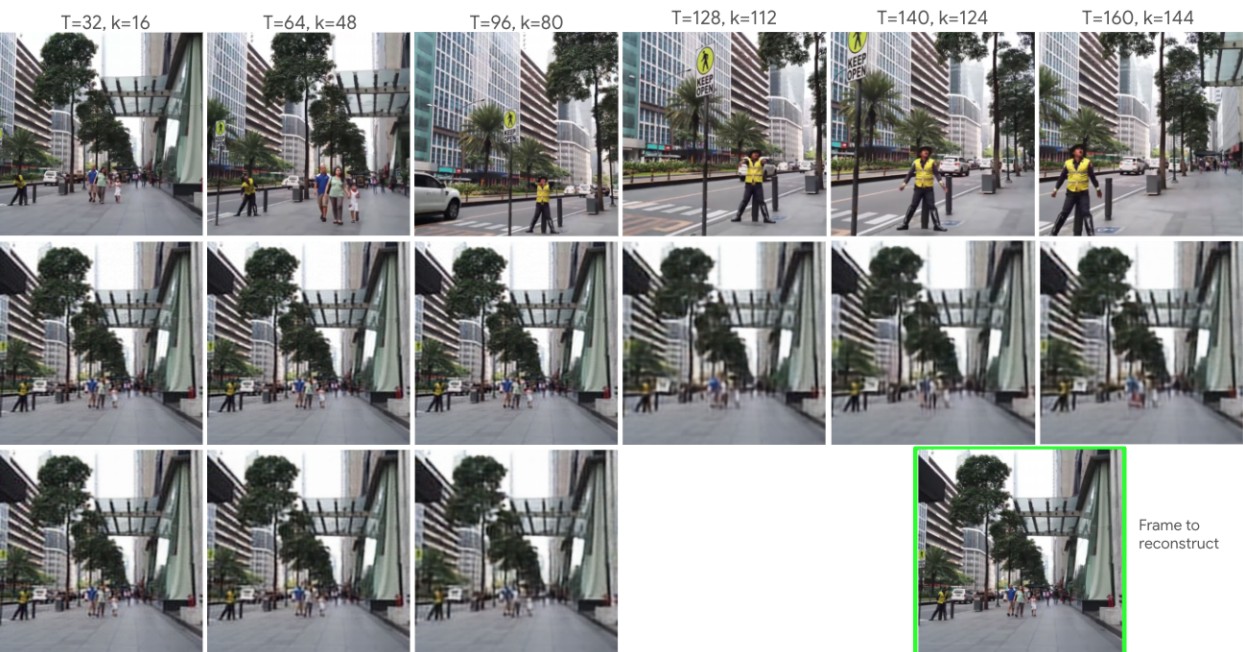

Figure 8: Qualitative results for the task of reconstructing a frame from the past, for increasing distance $k$ to the frame to reconstruct from left to right. **First row**: last frame seen by the model. **Second row**: TRecViT output. **Third row**: ViViT-L output; ViViT-L goes OOM for $k > 80$, so no predictions are shown.

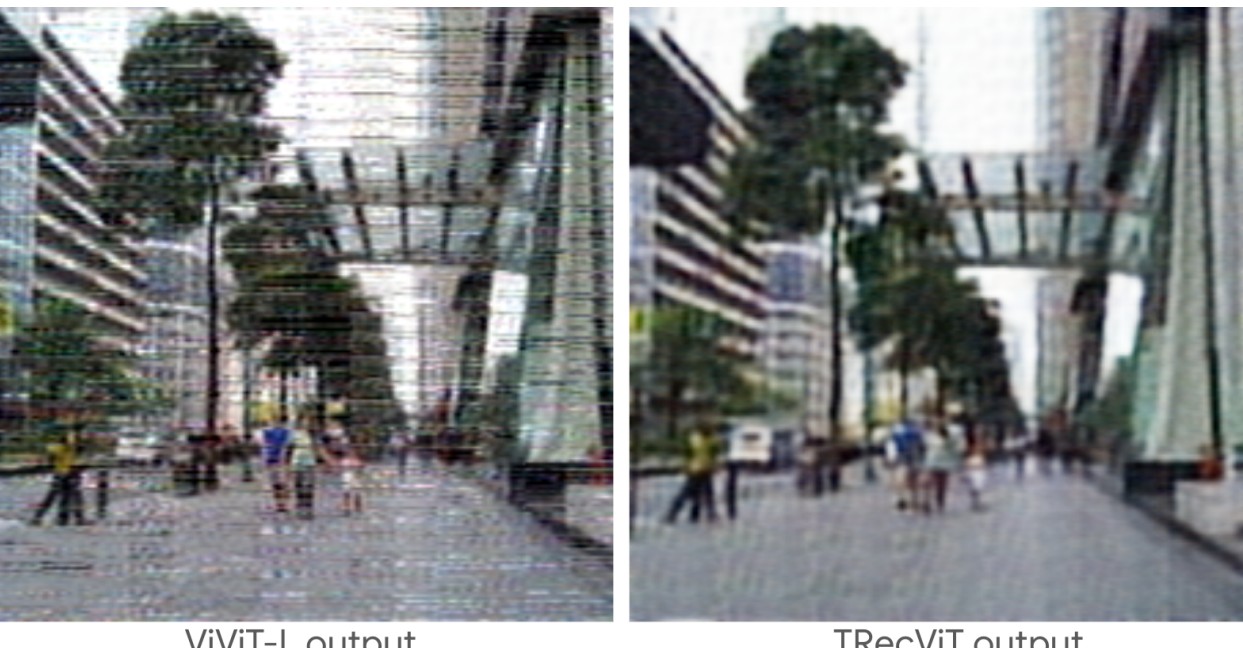

Figure 9: Generalisation to longer sequences. Both models are trained using Imagenet pre-trained weights, on video sequences of $T = 64$ frames to reconstruct the $16^{\text{th}}$ frame; during evaluation, the models receive sequences of $T = 96$ frames.