# OpenReview forum: "TRecViT: A Recurrent Video Transformer"
_TMLR — Accepted by TMLR_

### Review · Reviewer_GSRR · 2025-11-16

**Summary Of Contributions:**

This paper introduces TRecViT, a novel recurrent Transformer model for video modeling that works in both supervised or self-supervised settings. It leverages gated Linear Recurrent Units (LRUs) for temporal mixing, while employing ViT's self-attention mechanisms and MLPs for spatial and channel mixing, respectively. This TRecViT model is the only causal model within the state-space family within the video domain.  As a result, it achieves strong performance while using significantly less memory, fewer FLOPs, and reaching real-time inference speeds (~300 FPS). Overall, the model shows strong promise for efficient and scalable video applications.

Strengths:

1. TRecViT is a casual video model that combines SSM's recurrence (gated LRUs) with Transformer (ViT). Each framework is modeled by ViTs, while the recurrent state efficiently models temporal information. This hybrid design unifies SSM recurrence with Transformer structure, enabling to model temporal, spatial, and channel information.

2. The gated LRU provides data-dependent information flow with respect to recurrent and forget gating for updating hidden state. The constant C expands the effective forget range, allowing to flexibly model short-term and long-term temporal dependencies.

3. TTRecViT supports both supervised learning for label prediction and self-supervised learning via VideoMAE task for representation learning.

4. The memory footprint and the number of FLOPs of TRecViT against multiple baselines.

5. The paper includes extensive ablation studies that evaluate various component of the proposed method, clearly demonstrating the advantages of the architecture.

**Additional Comments:**

1. In introduction, it states "However, none of the existing proposed architectures can run in a causal manner, their performance strongly depends on bidirectional operation". Many established sequence models are designed for causal modeling. The manuscript should revise this statement to avoid over-claiming or more accurately describe the objects "the existing proposed architectures".

2. Fix the typo in page 6 "then we apply a 1D convolution followed by the gated LRU described in equation equation 4".

**Audience:**

Yes

**Audience Explanation:**

Yes. The paper addresses an important topic in video modeling by combining SSM's recurrence with ViT. It provides a novel way to model both temporal, spatial, channel information. The effective efficient architectures are highly relevant to TMLR’s audience.

**Broader Impact Concerns:**

This research focuses on developing efficient video modeling architectures. It does not directly involve sensitive data, human subjects, or high-risk deployment scenarios. It does not have broader impact concerns.

**Claims And Evidence:**

Yes

**Claims Explanation:**

Overall, most claims are supported by strong experiments, including baseline comparisons, ablations, and multi-task evaluations.

However, the efficiency claims are not fully demonstrated. This paper only includes the memory and runtime analysis based on its temporal block (gated LRU) rather than the overall model, which includes the memory and time consuming ViT components.

**Requested Changes:**

1. Move LRU Background Out of the Method Section.  Section 3.1 discusses the The background on LRUs, which is not the contribution of this paper and not part of the proposed method. I recommend moving the LRU background to the Related Work section, so that Section 3 focuses exclusively on the proposed architecture.

2. Add Clear Notation Before Referring to Figure 1. In page 3, Section 3 describes the model pipeline and refers to Figure 1, but the necessary notation (e.g., video data, temporal tube, spatial tokens) is only introduced much later in Section 3.2. Note that Figure 1 has variable notations, making readers must jump to Section 3.2 to find their meanings. Therefore, the notations are better introduced at the beginning of Section 3, maybe having a formal problem definition with variable notations.

3. Figure 3 only evaluates the gated LRU module, whereas we are interested i the overall efficiency of the entire architecture. Moreover, the ViT blocks contribute substantial cost. Please provide full-model memory and FLOPs comparisons, not just temporal module  gated LRU.

4. Lack of theoretical complexity analysis: The paper does not provide a theoretical analysis of the model’s computational complexity, making it difficult to clearly understand how the proposed architecture scales compared to RNNs or Transformers (e.g., ViViT).

5. Missing discussion of inherited limitations. Since the model combines recurrent units with Transformer components, it should have both advantages and disadvantages. As discussed in introduction, RNNs are slow to train due to their sequential processing, while Transformers suffer from quadratic complexity in self-attention.These limitations should be explicitly discussed for TRecViT.

6. The paper does not report essential architectural hyperparameters (e.g., embedding dimension, number of heads, number of layers, patch size for embedding). It only provides the model parameters.

---

> ### Author Response · Authors · 2025-12-05
> **Suggested changes to structure, clarifications on efficiency, discussion of complexity and limitations**
>
> We thank Reviewer GSRR for their in-depth review and for the very useful suggestions related to the structure of the method section, discussion of complexity and limitations. We address all the comments below. The quoted paragraphs have been included in the updated draft in blue.
>
> (1) **LRU background**: Thank you for the suggestion, we have moved this part under the Related work section to streamline the Method section.
>
> (2) **Notations**: Thank you for the suggestion. We have moved the notations at the beginning of section 3.
>
> (3) **Efficiency of the architecture**: We apologise for using a confusing notation in Figure 3. This has been updated now. The memory and FLOPs reported are for the full model (including the ViT blocks), not only for the gated LRU. The figure has been updated to compare against different variants of ViViT for clarity. The FLOPS and memory comparison for other hybrid baselines (ViT-LSTM, ViT-Conv1D) is now included in the appendix (Figure 5).
>
> (4) **Theoretical complexity**: Thank you for the suggestion. We have included a discussion on the complexity of the model in section 4.2.
>
> (5) **Limitations**: We added a paragraph on limitations under the Conclusion section.
> “*Limitations and Future Work: While this study presents the first exploration into leveraging LRUs for
> the video domain, some key aspects necessitate further investigation. First, we plan to conduct a deeper investigation into the memorisation limitations of our architecture, particularly in needle-in-haystack problems. Second, we aim to understand our model’s capabilities in a multi-step generative context, specifically by integrating it within video diffusion models. Finally, given LRUs' inspiration from continuous time systems, we plan to explore its capabilities of modelling variable frame rate videos across various complex applications, including video-language tasks and Robotics.*”
>
> (6) **Architecture hyperparameters**: Thank you for the comment. We are relying on the standard ViT block configurations. For clarity, we have now included all these configurations in the appendix, Table 8.
>
> **Minor comments:**
> (1)"the existing proposed architectures" --> We have now clarified that we are referring specifically to *existing video SSM architectures*. (2) Fixed typo "equation equation"

---

> ### Comment · Reviewer_GSRR · 2025-12-06
> **Updated Comment**
>
> Thanks for your update. I checked it, and it has addressed my concerns.

---

### Review · Reviewer_KCcQ · 2025-11-23

**Summary Of Contributions:**

The paper proposes a method that combines gated-LRU units for temporal modeling with ViT blocks for spatial modeling, resulting in an effective video representation learner. It also seems to be the first causal video model within the state-space–modeling framework. The extensive set of experiments demonstrates that the proposed approach is not only computationally efficient, but also outperforms several memory-intensive baselines such as ViViT. This highlights both the effectiveness and practicality of the method for large-scale video understanding.

**Audience:**

Yes

**Audience Explanation:**

1) The findings would be of interest to the TMLR audience. The work introduces an alternative to self-attention–based video models, offering significantly reduced computational and memory costs while still achieving competitive or superior performance. This opens new research directions in video modeling, especially in the context of state-space models and causal temporal architectures.

2) The method shows strong empirical performance in the self-supervised setting and even surpasses several supervised baselines on video classification tasks. This makes the proposed approach further usable or extendable for different video tasks.

**Broader Impact Concerns:**

As such, the paper itself does not pose any direct technological concerns. However, video modeling techniques in general could potentially be misused, for example, for surveillance without consent. That said, in this particular case, a broader impact statement may not be strictly necessary in my opinion

**Claims And Evidence:**

Yes

**Claims Explanation:**

1) The claims made in the paper are supported by clear, convincing, and comprehensive empirical evidence. The authors propose a new way of modeling videos through the combination of gated-LRU temporal modules and ViT-based spatial modeling. This design is both effective and computationally efficient, and the experiments consistently validate these claims across multiple benchmarks.

2) The paper is also clearly written and well organized, making the technical contributions accessible.

**Requested Changes:**

1) The paper mentions that the LRUs are "space-shared". Does this imply that LRU1 through LRU4 are actually the same module with shared parameters? If so, it is unclear why they are annotated differently in Figure 1. Clarifying this point in the paper would be very helpful.

2) The proposed architecture applies **temporal modeling first** using LRUs, followed by "spatial modeling" with ViT blocks. Is there a specific motivation for choosing this order? In principle, one could also apply spatial modeling first and then temporal modeling on top of the ViT-encoded representations. It would be useful for the paper to discuss why the chosen ordering is preferable.

3) The introduction suggests that this is the first work to perform "causal temporal modeling" within the state-space–modeling framework. However, the paper would benefit from a clearer explanation of how the gated-LRU implicitly enforces causal video modeling, and how this mechanism differs from prior approaches.

4) Would it be accurate to view this method as an extension of Griffin to the video domain? Although the underlying LRU building block follows the formulation proposed by De et al., the way temporal and spatial information are combined in this work goes beyond a straightforward adaptation. De et al. focus on language modeling with a Transformer-like architecture built from gated-LRUs, while this paper applies LRUs to sequences of visual tokens. It is not entirely clear whether parameters such as "C" behave identically when the input consists of visual rather than textual embeddings. Additional explanation, or even a small experiment, would help clarify whether the same parameterization generalizes appropriately across modalities.

5) Since LRUs model stable temporal dynamics, and the current self-supervised objective focuses on reconstructing the _current_ frame, it raises the question of what modifications would be required to enable "future-frame extrapolation". Understanding how well the LRU dynamics extend to multi-step prediction would be highly interesting.

6) Finally, it would be useful for the paper to discuss any "limitations" of the proposed approach. Highlighting scenarios where the method may underperform, or tasks for which LRUs may not be ideal, would help readers better understand when and where the model is most appropriate.

Minor:
1)  The text inside Figure 3 is quite small. Increasing the font size slightly would make it much easier to read.
2)  The paper does not clearly explain which evaluation metrics are used for each task. Adding a short section summarizing the metrics would help readers understand the comparisons.
3)  Section 5.4 refers to Figure 8, but this figure does not appear in the main paper. It seems to have been moved to the supplementary material. It would be better to include it in the main paper for clarity.

---

> ### Author Response · Authors · 2025-12-05
> **Clarifications on notation, order of operations, causality of our architecture, discussion of limitations**
>
> We would like to thank Reviewer KCcQ for their thorough and insightful review. The reviewer posed several interesting questions that we address in detail below. The quoted paragraphs have been included in the updated draft in blue.
>
> (1) **Notation LRU1, LRU2 ...**: We added this note at the beginning of section 3 to clarify the notation: “*… without mixing the information across temporal tubes, so each LRU has its own state – we use LRU1, LRU2,... in Figure 1 to highlight this.*”
>
> (2) **Order of operations**: Thank you for the question. We added a note in the paper to clarify this before section 3.1. “*We first perform temporal mixing, followed by spatial and channel mixing. We found this time-space order to produce better results compared to space-time order. We hypothesise that, by applying LRUs first, this allows them to focus on more local, easier to model, information at the first layer, instead of operating directly on features that mix information across the entire frame.*”
>
> (3) **causality of TRecViT**: This is a very good point. We updated a sentence in the introduction to clarify this:
> “*Importantly, by restricting the LRUs to temporal-only recurrence, this factorisation reduces the sequence length by about two orders of magnitude compared to models that apply recurrence across both space and time (Zhu et al., 2024; Li et al., 2024). Such models typically require bidirectional scanning to attain strong performance, preventing them from operating in a causal manner.*”
>
> (4) **Language tokens vs visual tokens**: Indeed, we are applying Griffin-like blocks in a setup that is very different from the original language setup, due to the different nature of the video signal and the proposed factorisation.
> For this reason, we ran the analysis on C’s eigenvalues initialisation in section 3.1 and the ablations in section 5.1. We added this sentence in section 3.1 to highlight this:
> “*Although LRUs have been proposed for language, we hypothesise that they can effectively model video signals, given the continuous nature of the video and LRUs' inspiration from time continuous systems. We run extensive analysis and ablations for the different components and hyperparameters of the gated LRU block to find a configuration that works well for video; see analysis below and ablations in section 5.1.*”
> This sentence was added in Conclusion as well:
> “*Finally, the LRUs’ unique ability to model continuous signals suggests promising application to tasks requiring reasoning about variable frame rates and frequency components within continuous video streams.*”
>
> (5) **Future-frame extrapolation**: This is an interesting question. From a practical perspective, learning good video representations using self-supervised learning like masked autoencoding, requires conditioning on some ground truth patches to avoid collapsing to the mean (e.g. as done in siamese models https://arxiv.org/abs/2305.14344, learning from one video stream https://arxiv.org/abs/2312.00598). If such conditioning is allowed, then predicting a future frame is similar to reconstructing the current frame. More generally, we agree that better understanding LRU dynamics for multi-step prediction, even in generative setups like diffusion, would be highly relevant. We included this as future work in Conclusion.
> “*we aim to understand our model’s capabilities in a multi-step generative context, specifically by integrating it within video diffusion models.*”
>
> (6) **Limitations**: We added a paragraph on limitations under the Conclusion section.
> “*Limitations and Future Work: While this study presents the first exploration into leveraging LRUs for
> the video domain, some key aspects necessitate further investigation. First, we plan to conduct a deeper investigation into the memorisation limitations of our architecture, particularly in needle-in-haystack problems. Second, we aim to understand our model’s capabilities in a multi-step generative context, specifically by integrating it within video diffusion models. Finally, given LRUs' inspiration from continuous time systems, we plan to explore its capabilities of modelling variable frame rate videos across various complex applications, including video-language tasks and Robotics.*”
>
> **Minor comments**:
> (1) Quality of fig 3: This has been updated.
> (2) Evaluation metrics: We added definitions of the metrics used in the relevant sections, the first time when they are mentioned in the text.
> (3) Figure in the appendix: We clarified that the qualitative results are included in the appendix (page 12): “see qualitative results in the appendix, Figures 8 and 9”. Given the size of these figures, we think that they can be better visualised in the appendix, where we don’t have page limit constraints.

---

> > ### Comment · Reviewer_KCcQ · 2025-12-08
> > **Updated Comments**
> >
> > The updated version and the additional explanations fully addressed my concern. Thank you.

---

### Review · Reviewer_xPAN · 2025-11-23

**Summary Of Contributions:**

The paper proposes TRecViT, a causal video architecture for video understanding. Its core contribution is a new temporal module that alternates gated LRUs along the temporal dimension, yielding a causal architecture that can operate in streaming or robotics oriented settings. The authors present TRecViT as the first LRU based video model and provide detailed analyses showing large efficiency gains relative to ViViT. They further demonstrate versatility across tasks and training regimes, including supervised classification on SSv2 and K400, MAE style self-supervised pretraining, and point tracking.

On the positive side, the architecture is causal and efficient, achieving strong throughput while delivering strong performance with relatively few parameters compared to transformer baselines, and its ability to support video classification, video prediction, and dense point tracking indicates that it can serve as a general purpose video backbone.

On the negative side, the novelty may be perceived as largely an architectural recombination of existing components, namely ViT style spatial transformers and gated LRUs, rather than a new modeling principle. The results on Kinetics 400 are mostly competitive rather than clearly superior to strong transformer baselines.

**Audience:**

Yes

**Audience Explanation:**

Video understanding is an important and active research area, and the design of video architectures is definitely an important topic within it. This paper proposes a new architecture and validates it with extensive experiments, so its findings are very likely to be of interest to the TMLR audience.

**Claims And Evidence:**

Yes

**Claims Explanation:**

The paper’s main claims that the proposed time–space factorization with LRUs yields a causal video backbone, and that this backbone achieves state-of-the-art or competitive performance against both causal and non-causal baselines under comparable settings, are supported by extensive quantitative experiments on SSv2 and K400.

**Requested Changes:**

1. Although the temporal shift operation in TSM supports an online setting, it can essentially be viewed as a channel-wise 1D convolution with fixed kernel weights, so I am not fully convinced that it should be regarded as causal model. It would be helpful if the authors could clarify this point.

2. I also find the comparison between TRecViT and ViViT in terms of FLOPs and parameter counts somewhat overclaimed or potentially misleading. TRecViT appears to be designed on top of a ViViT-B, or more precisely ViT-B backbone, so it should primarily be compared against ViViT-B rather than ViViT-L. Moreover, since TRecViT introduces additional modules, it naturally ends up with more parameters than ViViT-B, which should be acknowledged. Finally, TRecViT is consistently compared against the full self-attention variant of ViViT. For fair comparison, it would be more appropriate to evaluate it against the factorized variant of ViViT (ViViT-FE Base).

---

> ### Author Response · Authors · 2025-12-05
> **TSM as a causal model and comparison to ViViT**
>
> We thank Reviewer xPAN for their insightful review. The reviewer raised concerns about (1) the causal nature of TSM model, (2) the comparison against ViViT, and (3) the novelty of the architecture.
>
> We address these concerns below. The quoted paragraphs have been included in the updated draft (in blue).
>
> (1) **TSM as causal model**: As detailed in the TSM paper https://arxiv.org/pdf/1811.08383 section 4.2 and Figure 4, in the online setup, the feature shift happens only from previous frames to the current frame, resulting in a uni-directional TSM. Since the model doesn’t have access to information from future frames, we considered it causal. The original TSM paper does not report the online accuracy on SSv2 dataset (reports only on SSv1 – version which has been discontinued in the meantime due to label noise). We borrowed the online accuracy for TSM on SSv2 from the RViT paper, Table 3, https://openaccess.thecvf.com/content/CVPR2022/papers/Yang_Recurring_the_Transformer_for_Video_Action_Recognition_CVPR_2022_paper.pdf. We added a note in the paper to clarify the source of this result (Table 3 caption).
>
> (2) **Comparison against ViViT**: We thank the reviewer for raising this point that could indeed be a source of confusion. We added a clarification in the updated section on Computational complexity (section 4.2) and we included ViViT-L factorised encoder in Figure 3 together with explanation in the caption of Figure 3.
>
> “*We discuss the computational complexity of the proposed architecture and its efficiency compared to different variants of ViViT. Our model's parameter count (111M) falls between that of ViViT-B (90M) and ViViT-L (310M). We focus on ViViT-L as the main point of comparison in our efficiency and SOTA analysis (Section 5.2) as this is the strongest representative of the ViViT family reported in the original ViViT paper. Furthermore, since our architecture is suitable for both sparse and dense video tasks, we compare the efficiency benefits mainly against the more general full self-attention ViViT variant, which is utilised by influential follow-up works (e.g., VideoMAE) over the specialised Factorised Encoder (FE) version, which models temporally only one token per frame, being suitable for classification tasks, but not for dense tasks.*"
>
> (3) **Novelty**: We added a couple of sentences that attempt to address the novelty concern:
>
> - The proposed factorisation is novel and enables our model to run in a causal manner, being the first model in the SSM family to run causally;
> sentence added in introduction: “*Importantly, by restricting the LRUs to temporal-only recurrence, this factorization reduces the sequence length by about two orders of magnitude compared to models that apply recurrence across both space and time (Zhu et al., 2024; Li et al., 2024). Such models typically require bidirectional scanning to attain strong performance, preventing them from operating in a causal manner.*”
>
> - LRUs have been designed for language sequences. Video is a significantly different modality, continuous in nature, with spatial dimensions in addition to time. Our factorisation separates space from time and allows us to apply LRUs on video in an efficient manner. Because this is a different modality, we ran analysis and ablations to study the role of each component within gated LRUs, and find optimal initialisation values;
> sentence added in section 3.1: “*Although LRUs have been proposed for language, we hypothesise that they can efficiently model video signals, given the continuous nature of the video and LRUs' inspiration from time continuous systems. We run extensive analysis and ablations for the different components and hyperparameters of the gated LRU block to find a configuration that works well for video; see analysis below and ablations in section 5.1*”

---

> ### Comment · Reviewer_xPAN · 2025-12-13
> **Response to Authors**
>
> Thanks to the authors for the clarification. My concerns have been adequately addressed.

---

### Author Response · Authors · 2025-12-05
**Answer to reviews**

We would like to deeply thank the reviewers for their time and attention to review our manuscript in detail and provide insightful and constructive feedback. The reviewers appreciated that this is an efficient and versatile causal architecture (xPAN, KCcQ), achieving strong performance as demonstrated by extensive experiments (xPAN, KCcQ, GSRR), and that the paper is clearly written (KCcQ).

The main comments were around the lack of limitations discussion (KCcQ, GSRR), lack of theoretical complexity discussion (GSRR), comparison against ViViT and TSM baselines (xPAN), and clarifications around some design choices (KCcQ).

We have addressed all these points in individual answers to reviewers and we have added clarifications for all the raised points in the updated draft (marked in blue).

---

### Decision · Action_Editor_ANeD · 2025-12-23

**Recommendation:** Accept as is

**Audience:**

Yes

**Audience Explanation:**

The paper proposes a video backbone based on state-space framework which is an interesting avenue of research.

**Claims And Evidence:**

Yes

**Claims Explanation:**

The paper proposes an approach for causal video modeling based on a state-space framework.  The paper has strong empirical performance in classification, video prediction, and point tracking tasks.  In the revised version, the authors clarified the reviewers' concerns and improved the presentation.  Given the methodological contribution and the strong empirical results, the reviewers recommend the paper's acceptance.